# Iron-Induced Liver Injury: A Critical Reappraisal

**DOI:** 10.3390/ijms20092132

**Published:** 2019-04-30

**Authors:** Steven A. Bloomer, Kyle E. Brown

**Affiliations:** 1Division of Science and Engineering, Penn State University, Abington College, Abington, PA 19001, USA; sab320@psu.edu; 2Iowa City Veterans Administration Medical Center, Iowa City, IA 52242, USA; 3Division of Gastroenterology-Hepatology, University of Iowa Roy J. and Lucille A. Carver College of Medicine, Iowa City, IA 52242, USA; 4Program in Free Radical and Radiation Biology, University of Iowa Roy J. and Lucille A. Carver College of Medicine, Iowa City, IA 52242, USA

**Keywords:** chelation, cirrhosis, hemochromatosis, iron, liver injury, phlebotomy

## Abstract

Iron is implicated in the pathogenesis of a number of human liver diseases. Hereditary hemochromatosis is the classical example of a liver disease caused by iron, but iron is commonly believed to contribute to the progression of other forms of chronic liver disease such as hepatitis C infection and nonalcoholic fatty liver disease. In this review, we present data from cell culture experiments, animal models, and clinical studies that address the hepatotoxicity of iron. These data demonstrate that iron overload is only weakly fibrogenic in animal models and rarely causes serious liver damage in humans, calling into question the concept that iron overload is an important cause of hepatotoxicity. In situations where iron is pathogenic, iron-induced liver damage may be potentiated by coexisting inflammation, with the resulting hepatocyte necrosis an important factor driving the fibrogenic response. Based on the foregoing evidence that iron is less hepatotoxic than is generally assumed, claims that assign a causal role to iron in liver injury in either animal models or human liver disease should be carefully evaluated.

## 1. Introduction

The presumption that iron-catalyzed oxidative injury plays a key role in the pathology of various forms of liver disease permeates the scientific and clinical literature. On the one hand, this notion is supported by evidence from cell culture models demonstrating cytotoxicity in response to iron (often at supraphysiological concentrations) and, on the other, by associations between common forms of chronic liver disease and secondary abnormalities of iron metabolism that are widely viewed as contributing to disease pathogenesis. In this review, we present findings from animal models as well as clinical data that challenge the notion that iron overload is hepatotoxic in vivo. We have focused on iron as a causative agent in chronic liver injury manifested by fibrosis and/or cirrhosis in animal models and in humans. The association of hepatic iron deposition with hepatocellular carcinoma (HCC) is an important topic, but one that is beyond the scope of this review. Readers interested in this topic are referred to recent reviews [1,2,3].

## 2. Role of the Liver in Whole-Body Iron Regulation

The liver plays a major role in whole-body iron metabolism. It is a major storage site for iron, it takes up transferrin- and non-transferrin-bound iron, and releases iron back into the circulation. Additionally, the liver is a sensor of whole-body iron stores, regulating intestinal absorption of dietary iron and the release of recycled iron from macrophages in response to multiple physiologic factors. A key regulator of these processes is the hormone hepcidin, which is produced by hepatocytes. Hepcidin levels increase in response to elevated transferrin saturation or inflammatory cytokines, which results in decreased uptake of iron across the intestinal epithelium and diminished release of iron from macrophages. The receptor for hepcidin, ferroportin, is the only known mammalian iron exporter, and its degradation after binding hepcidin prevents iron efflux [4]. Inflammatory stimuli have been shown to downregulate expression of ferroportin, indicating that it is a negative acute phase protein [5,6]. Interestingly, ferroportin has been localized to the nucleus, where it may have an important role in intracellular iron trafficking at times when hepatocytes have increased metabolic activity [6]. Decreased levels of iron diminish the expression of hepcidin, as does enhanced erythropoietic drive. In the latter case, erythropoietin-stimulated erythroferrone production suppresses hepatic expression of hepcidin [7]. Both pathways are associated with decreased levels of hepcidin, which abrogates its effects on intestinal and macrophage iron handling. Dysregulation of hepcidin is presumed to be a common denominator in conditions characterized by iron overload. As this review is concerned primarily with the effects of excess iron on the liver, readers desiring additional details about the physiology of hepcidin are referred to review articles on this topic [8].

## 3. Iron Uptake and Storage in Hepatic Cells

Hepatocytes are rich in heme-containing enzymes such as catalase and the cytochromes p450; thus, iron is required for their synthesis. The major route of entry of iron into hepatocytes is via the classical system of receptor-mediated endocytosis of diferric transferrin (Tf) by transferrin receptor-1 (TFR1), which is common to all cells in the body. After endocytosis of transferrin-bound iron-TFR1 complex, the iron is released from Tf in the acidic lysosomal compartment, which is an important subcellular storage area for iron due to its acidic pH (iron is soluble at low pH). The iron in the lysosomal compartment can be delivered to other organelles, such as the mitochondrion, for incorporation into macromolecules containing heme or iron-sulfur centers. Iron in excess of cellular needs is oxidized to the ferric form, and sequestered within the iron storage protein, ferritin, where it is kept in a relatively inert state. When cellular iron stores exceed ferritin synthesis and storage, iron enters a secondary storage site, hemosiderin, which is a membrane-enclosed structure that contains ferric iron and resembles a residual body of a lysosome. When systemic demand for iron increases (e.g., due to enhanced erythropoiesis), storage iron is mobilized and exported from hepatocytes via ferroportin, entering the circulation as diferric transferrin. Because there is no regulated means for excreting iron, disturbances in the mechanisms that regulate iron uptake can cause excess iron deposition in the liver and other organs such as the pancreas and the heart.

In addition to iron uptake using the classical Tf-TFR1 system, hepatocytes also take up non-transferrin bound iron (NTBI) through dimetal ion transporter-1 (DMT1) [9] and Slc39a14 [10]; this iron can then be stored or delivered to various organelles. Significant amounts of NTBI occur in conditions such as hereditary hemochromatosis and studies in isolated perfused rat livers have shown that hepatocyte uptake of NTBI is both highly efficient and unaffected by hepatic iron stores, suggesting that the liver is the preferred site of clearance of NTBI [11,12]. By simultaneously tracking NTBI and hepatic iron stores, investigations also suggest that the liver clears plasma iron after inflammatory and necrotic stimuli [6,13]. In addition, the liver participates in minimizing the potential toxicity of circulating heme and hemoglobin via the synthesis of hemopexin and haptoglobin, plasma proteins that bind heme and hemoglobin, respectively. Internalization of the complexes formed between the heme-containing molecule and its binding protein are mediated by specific receptors for hemopexin on hepatocytes and macrophages and for haptoglobin, which are found predominately on hepatic macrophages (Kupffer cells). Along with the macrophages of the spleen, Kupffer cells phagocytose effete red blood cells, and degrade the heme molecules to biliverdin, carbon monoxide, and iron via the enzyme, heme oxygenase. The iron liberated in this reaction is incorporated into ferritin [14]. Thus, the liver plays a major role in protecting other organs from circulating iron and heme.

## 4. Iron-Induced Cellular Injury: In Vitro and In Vivo Models

It is widely assumed that an important mechanism for the toxicity of iron derives from the ability of ferrous iron to interact with H_2_O_2_ to generate the highly reactive hydroxyl radical, i.e., the Fenton reaction. Since H_2_O_2_ is a ubiquitous product of cellular metabolism, iron that is not tightly bound to ferritin or hemosiderin (sometimes called “free” or labile iron) may participate in reactions that generate this highly reactive and highly damaging species in biological systems. As a consequence, under physiological circumstances, tight regulation of intracellular iron plays an important role in preventing oxidative damage to macromolecules such as proteins, lipids, and nucleic acids [15]. It is important to note that while the Fenton reaction has been studied extensively in the test tube, direct evidence for the formation of hydroxyl radicals in intact tissues or organs of iron-loaded animals is limited. Using electron paramagnetic resonance spectroscopy, Figueiredo and colleagues were able to demonstrate hydroxyl radicals in lysates of iron-loaded rat livers processed at pH 5.0 but not in lysates processed at pH 7.4. The authors suggested that these findings might support a role for iron-laden lysosomes in iron-induced liver injury [16]. Using a similar approach, Kadiiska and co-workers observed hydroxyl radicals in the bile of rats fed a diet high in ferric citrate for 10 weeks [17]. Interestingly, although ferric citrate treatment increased serum iron concentrations, there was no demonstrable increase in hepatic iron concentrations, hence, strictly speaking, this is not a model of iron overload. 

Despite the paucity of evidence for direct involvement of hydroxyl radicals, iron-mediated injury has been described in many cell types. In this section, we will limit our discussion to hepatocytes and the liver. Several investigations have demonstrated a causal role for iron in H_2_O_2_-mediated necrosis in hepatocytes. In those studies, iron sensitized cells to death while the lysosomal iron chelator, deferoxamine (DFO), prevented cytotoxicity [18,19,20]. While robust effects of DFO were demonstrated, the concentrations of H_2_O_2_ used in these experiments, given exogenously in millimolar concentrations or generated via the glucose–glucose oxidase system, were quite high (0.25–2 mM) relative to physiologic H_2_O_2_ concentrations. Cells in vivo are rarely exposed to extracellular concentrations of H_2_O_2_ exceeding 100 µM [21]. Extracellular H_2_O_2_ concentrations of 5–7 µM result in intracellular H_2_O_2_ concentrations in the range of 7.5–10 nM, which are sufficient to initiate apoptosis [22]. Addition of iron salts to cultured hepatocytes also has been shown to cause an increase in lipid peroxidation [23,24]. However, like the peroxide studies, the amounts of iron used in these in vitro studies (0.1 mM) are much higher than normal plasma iron concentrations (≈0.018 mM or 100 µg/dL). Even under conditions of pathological iron overload, most iron is bound to transferrin, ferritin, etc., with relatively small amounts present as NTBI. Not only were the concentrations of iron supraphysiologic in these studies, but the form in which iron was administered (i.e., as an iron salt) ensured that the cells were exposed to much higher levels of what is effectively free iron than is likely to occur even under pathophysiological conditions. These results demonstrate the potential for interactions between iron and H_2_O_2_ to cause cellular injury in some circumstances, but emphasize that the conditions required for these interactions rarely, if ever, are present in intact animals, limiting their relevance to naturally occurring forms of liver injury. 

An exception to the limited toxicity of iron may be in the context of combined exposure with other toxins. Using more physiological concentrations of iron (0–10 µM of trimethyl-hexanoyl ferrocene; TMHF), Moon and associates demonstrated that iron-induced cytotoxicity was potentiated by treatment with acetaminophen and vice-versa [25]. TMHF alone was associated with mild cytotoxicity in mouse primary hepatocytes only at the highest concentration tested; however, when combined with 10 mM acetaminophen, cytotoxicity was observed with 5 µM TMHF. Conversely, the cytotoxic dose of acetaminophen was lowered from 20 mM to 5 mM in the presence of 10 µM TMHF. These results suggest that iron can potentiate injury when other toxins or injurious processes are present, a concept that will be further elaborated upon in the in vivo studies below. 

Similarly, various cellular and physiological perturbations have been shown to alter iron homeostasis, as well as the expression of iron regulatory genes [5,26]. In concert with exogenous iron or oxidants, these alterations in iron handling may potentiate cellular injury. In primary hepatocytes, nutrient deprivation-induced autophagy increased the chelatable or labile iron pool [27]. Treatment of these cells with ferric chloride sensitized the cells to cell death, while DFO completely protected against cell death [27]. In response to bafilomycin and *tert*-butyl hydroperoxide, labile iron was released from lysosomes into the cytosol and mitochondria of mouse hepatocytes [28]. Treatment of these cells with DFO decreased cytotoxicity, suggesting a causal role of lysosomal iron liberation in cell death [28]. Some in vivo models of oxidative injury also demonstrate a causal role of iron. Elevations in labile iron and lipid peroxidation have been demonstrated during heat stress in vivo, and during hyperthermic perfusion of the liver [29,30]. In these investigations, DFO decreased hyperthermia-induced lipid peroxidation, further implicating iron in the generation of oxidative injury in these models [29,30]. 

Dietary modifications have been shown to alter iron homeostasis in the liver in some animal models. In rabbits fed a high-fat diet (HFD), Otogawa and associates demonstrated an increase in hepatic iron, histological fibrosis (as assessed by Sirius red), the hepatic stellate cell activation marker, α-smooth muscle actin (α-SMA), and hydroxyproline, as well as an increase in the mRNA expression of transforming growth factor- β1 (TGFβ1), collagen α1(I), and pro-inflammatory cytokines [31]. HFD-induced iron deposition was reduced via phlebotomy, which was associated with a reduction in Sirius red staining and α-SMA, suggesting that iron plays a causal role in fibrosis in this model [31]. 

However, increases in iron are not an invariant feature of dietary models of hepatic injury. In a recent study, we demonstrated that a high-fat diet that produced histologic features of nonalcoholic steatohepatitis in mice was accompanied by increased markers of inflammation and oxidative stress and decreased hepatic iron concentrations [32]. Similar results were reported by Padda et al., who also observed a significant decrease in hepatic iron content with a high-fat diet [33]. Other studies have evaluated hepatic iron content in the context of caloric restriction in aging models. While one study in rats demonstrated an increase in hepatic iron with aging, and a decrease with caloric restriction [34], a study in mice demonstrated the opposite result [35]. Thus, it is far from clear that iron plays a causal role in oxidative stress or inflammation in these models. 

## 5. Iron and Inflammation

Iron has been implicated in promoting hepatic inflammation. A well-studied mechanism linking iron to inflammation involves the effects of iron on nuclear factor-κB (NF-κB) in Kupffer cells. NF-κB is a redox-sensitive transcription factor that stimulates the transcription of pro-inflammatory cytokines such as tumor necrosis factor-alpha (TNFα) and interleukin-6 (IL-6). In quiescent cells, NF-κB is bound to IκB, which sequesters this protein complex in the cytosol; however, oxidative stress activates IκB-kinase, leading to phosphorylation of IκB, which allows NF-κB to translocate to the nucleus [36]. Evidence in vitro and in vivo supports a role for iron in activating NF-κB. For example, in isolated Kupffer cells, iron administration stimulated NF-κB activity and increased concentrations of TNFα [37]. Moreover, the iron chelator, L1, suppressed lipopolysaccharide (LPS)-induced NF-κB activity as well as the induction of TNFα and IL-6 in Kupffer cells isolated from bile duct-ligated rats [38]. In vivo, chronic iron administration was associated with nearly 3-fold increases in hepatic TNFα mRNA in mice and rats [39,40]. Iron loading in mice also increased mRNA expression of other proinflammatory cytokines such as IL-6 and IL-1β [40]. These observations suggest that iron may modulate inflammation in various forms of liver injury; however, it is worth noting in this context that iron overload per se is usually not accompanied by a conspicuous inflammatory response (in the sense of histologically demonstrable infiltration of the liver with inflammatory cells), either in humans or most models of iron overload in rodents.

## 6. Iron and Hepatic Stellate Cells

Along with several other mesenchymal cell types [41], hepatic stellate cells (HSCs) play an important role in liver fibrosis. Since fibrosis is the final common pathway of many forms of chronic liver disease, it is important to consider the effects of iron on these pro-fibrogenic cells. Under normal conditions, HSCs are quiescent and store vitamin A, but when activated, they differentiate into myofibroblasts and produce collagen, thus leading to matrix deposition. In contrast to culture-activated myofibroblasts, quiescent HSC appear not to express transferrin receptor, suggesting minimal iron requirements under basal conditions [42]. A series of cell culture experiments demonstrated a role for iron in matrix production by cultured HSC. In particular, iron and transferrin increased αSMA expression and collagen synthesis [42,43,44], and these effects were blocked by DFO [43,44]. Although iron is required for several steps in the post-translational processing of collagen, this does not provide an obvious explanation for the effects of iron on αSMA expression. Results from Jin et al. confirm the above results and demonstrated that DFO also causes apoptosis in cultured HSCs [45]. The effect of iron on fibrogenic liver cell populations other than HSC is an important area for future research. 

In vivo models of iron overload seem to implicate iron in HSC activation. For example, chronic iron overload in rats increased the number of α-SMA-positive myofibroblasts and was associated with mild fibrosis [46]. In humans with hemochromatosis, a correlation was observed between hepatic iron concentration and numbers of αSMA-positive cells [47]. Interestingly, these authors demonstrated αSMA+ cells in the pericentral regions of the lobule, which is the inverse location of both iron deposition and of early fibrosis deposition in hemochromatosis, suggesting that the effects of iron on HSC αSMA expression in that condition are indirect. 

While iron seems to stimulate a pro-fibrogenic phenotype in HSCs in culture, HSCs activated in the setting of iron overload do not appear to accumulate iron themselves. In a model of iron overload in gerbils, ferritin was detected in hepatocytes, but not in collagen mRNA-positive HSCs, indicating that iron stores were not increased in the latter cell type, which again suggests an indirect effect of iron loading on HSCs [48]. To evaluate potential changes in intracellular labile iron, the authors also measured iron regulatory protein (IRP) activity, which is exquisitely sensitive to fluctuations in iron [49]. They demonstrated that IRP activity in HSCs isolated from iron-loaded gerbils was not different than HSCs isolated from control animals [48]. The authors speculated that paracrine factors released from damaged cells could be responsible for the activation of HSCs, in keeping with the general concept that injured hepatocytes are a source of cytokines, signaling molecules, etc., that may activate HSC directly [50,51]. Therefore, while iron can induce HSC activation in culture, the development of progressive fibrosis in vivo appears to require other factors (Figure 1).

## 7. Iron-Induced Liver Injury: Observations in Animal Models

Various animal models of iron overload have been developed to study the mechanisms by which iron causes hepatic fibrosis and cirrhosis. Importantly, most animal models of chronic iron overload, whether genetically mediated or achieved via administration of exogenous iron, fail to develop significant fibrosis when iron is the sole stressor [33,46,52,53,54,55,56,57,58,59,60,61]. Results of several key studies are summarized in Table 1. A variety of methods are used to produce iron overload. Enteral iron overload using diets enriched in various forms of iron, such as a carbonyl iron or ferrocene, are commonly used. In these dietary models, iron accumulates primarily in periportal hepatocytes during the early phase of iron loading, yielding a distribution of excess iron that closely mimics that of hereditary hemochromatosis (HH) [62,63]. In contrast, parenteral models of iron loading utilize compounds such as ferric nitrilotriacetate, saccharated iron, iron sorbitol or iron dextran, which can be administered intravenously, intramuscularly or intraperitoneally. Iron in this form is initially taken up by macrophages, but with repeated administration, iron is deposited within hepatocytes without a zonal pattern. This method produces a distribution of iron similar to that seen in secondary iron overload resulting from conditions such as thalassemia. Depending on the dose and duration of iron administration, these models can achieve upwards of 75-fold increases in hepatic iron concentrations in experimental animals without overt toxicity. 

Like the cell culture models, iron administration in vivo has been shown to result in increased levels of oxidative damage in the liver. Oxidative damage to lipids (i.e., lipid peroxidation, commonly measured using assays that quantitate thiobarbituric acid-reactive substances (TBARS), conjugated dienes, or specific lipid peroxidation products such as malondialdehyde, 4-hydroxynonenal or F2-isoprostanes) is the most frequently assessed parameter, although some studies have examined oxidative damage to proteins (e.g., protein carbonyls) and/or nucleic acids (such as 8-hydroxy-2-deoxyguanosine). As a rule, the extent of macromolecular damage in models of iron overload is relatively modest in comparison to the magnitude of iron overload [54,62,64]. In this context, it should be noted that some assays of oxidative damage are susceptible to artefactual generation of damage markers during sample preparation and the assay procedure [65]. This is a particular concern with samples that contain large amounts of iron. Iron-laden samples that are processed without taking precautions to prevent this (typically accomplished by the addition of an iron chelator with or without an additional antioxidant during sample preparation) may therefore overestimate the magnitude of oxidative damage occurring in vivo. 

The fibrogenic response in iron-loaded rodent livers tends to be even more modest than the increase in oxidative damage, with the typical iron loading experiment yielding small increases in hydroxyproline concentrations and collagen mRNA expression in the absence of significant histological fibrosis [46,54,57]. Although one might expect longer treatment or higher iron concentrations to potentiate fibrosis, this has generally not been the case. In an experiment using carbonyl iron-fed rats, we observed stable ≈2-fold increases in hepatic hydroxyproline from 4 through 14 months despite progressive increases in liver iron concentrations up to 30 times control levels over the same time course [46]. A 6-month experiment using biweekly injections of iron dextran achieved much higher hepatic iron concentrations (80–90-fold higher than control) with comparable effects on hydroxyproline levels as the carbonyl iron study [54]. 

There are some notable exceptions to the general rule that hepatic iron overload resulting from administration of exogenous iron fails to incite a fibrogenic response in rodents. Park et al. reported that a 12-month iron-loading protocol (3% dietary carbonyl iron) resulted in a 54-fold increase in hepatic iron concentrations and produced mild-to-moderate fibrosis in rats, which was associated with mild hepatocyte necrosis and leukocyte infiltration [66]. These results are difficult to reconcile with results by the same group of investigators showing no significant fibrosis after 14 months of iron loading using a similar protocol [46]. Valerio and Petersen reported a 15-fold increase in hepatic iron in C57Bl/6 mice fed a diet containing ferrocene for 4 months. This regimen produced mixed hepatocellular and sinusoidal lining cell iron deposition with large siderotic nodules preferentially located in centrilobular region of the liver. There was histologically-evident fibrosis in the livers of the iron-treated mice that was predominantly localized to the siderotic nodules [67]. This pattern of fibrosis, with collagen fibers in close proximity to clusters of iron-laden cells, is also seen with long-term carbonyl iron feeding [46]. It is worth noting that the large aggregations of iron-loaded cells seen in these models do not have an obvious counterpart in human HH and are not necessarily the equivalent of sideronecrosis, which is a more subtle finding that has been linked to fibrosis in HH [68]. Other studies have demonstrated the appearance of collagen fibrils after iron overload via electron microscopy (EM), although the necessity of EM to make this observation suggests that the extent of fibrosis in these studies was minimal [69,70,71]. In a recent study, Das et al. observed increases in histological fibrosis (via Sirius red and trichrome), procollagen mRNAs, and TGFβ mRNA after chronic iron loading in C57BL/6 mice [40]. Perhaps relevant to these results is the observation that the C57BL/6 strain has a propensity to develop inflammatory responses [72]. In this experiment, iron overload was accompanied by significant inflammation, as demonstrated by increased expression of IL-6, IL-1β, and TNFα mRNAs [40]. The induction of inflammation may be important in potentiating hepatic fibrosis in this model. 

The importance of inflammation as a cofactor driving fibrogenesis in rodent models of iron overload is further supported by investigations that utilized gerbils as the model organism [73,74]. While gerbils rapidly developed liver fibrosis after parenteral iron administration, this pathology developed on a background of pre-existing hepatic inflammation due to chronic endotoxinemia [73,74]. 

Genetic models of hepatic iron overload (resulting in hepatic iron concentrations 2 to 14 times normal) have likewise generally failed to produce robust hepatic fibrosis [75,76]. Examples include *HFE* knockout mice [60], hemojuvelin (Hjv) knockout mice [33,40], and hepcidin knockout mice [55]. Tan and colleagues found that expression levels of *Col1a1* and α-SMA did not differ between wild-type and *HFE* knockout mice, nor were there differences in the inflammatory markers, TNFα, or monocyte chemoattractant protein-1 (MCP-1) [60]. Alanine aminotransferase (ALT) and aspartate aminotransferase (AST) were elevated in Hjv knockout mice compared to wild type animals, but histological fibrosis was not detected, nor were there differences in TNFα, α1-(I)-collagen mRNA, or αSMA mRNA expression [33]. Lunova et al. found no histological fibrosis and no increase in hydroxyproline or collagen mRNA in the livers of hepcidin knockout mice versus wild type mice [55]. 

Similar to models of exogenous iron loading, genetic models of iron overload suggest that inflammatory responses may be an important modulator of the fibrogenic response to iron. The importance of this relationship is reinforced by findings from a study of mice with genetic alterations in two major regulators of iron metabolism, *HFE* and *TFR2* [77]. In this work, hepatic iron concentrations were increased about 3-fold versus controls in *HFE* knockout mice (*HFE*^−/−^) or mice with a mutation in *TFR2* (*TFR2^mut^*), and were similar to iron levels achieved by feeding a carbonyl iron-supplemented diet to wild-type mice. In the *HFE* knockouts and *TRF2* mutants, hydroxyproline levels and Sirius red quantification were elevated above control values, and both were similar to the iron-treated wild-type animals. The combination of the genetic alterations (*HFE*^−/−^
*xTfr2^mut^*) resulted in mice with greater increases in hepatic iron, markers of oxidative stress, and fibrosis (via both Sirius red and trichrome) compared to mice with either of the single genetic alterations and to iron-loaded wild-type mice. Foci of inflammatory cells were seen only in the livers of the *HFE*
^−/−^
*xTfr2^mut^* mice, which was attributed to hepatocyte sideronecrosis [77]. 

Broadly similar findings were observed in hepcidin knockout mice fed a 3% carbonyl iron diet for 12 months [55]. These mice showed mild increases in hepatic fibrosis compared to hepcidin knockouts fed a normal diet and to carbonyl iron-loaded wild-type mice. Liver iron content was approximately 3-fold higher in the iron-fed hepcidin knockouts compared to both of those control groups. The iron-loaded hepcidin knockouts also manifested histological inflammation and increased inflammation-related gene expression, suggesting a synergy between excess iron and inflammation driving fibrosis [55]. 

The means by which these genetic alterations +/− exogenous iron lead to sideronecrosis, inflammation, and ultimately fibrosis is an intriguing question. It seems doubtful that this is solely the result of the higher liver iron content in the *HFE*^−/−^
*xTfr2^mut^* or the iron-fed hepcidin knockout mice, given that other experimental protocols achieve high iron levels but nonetheless fail to generate an obvious inflammatory or fibrotic response. A recent study from Duarte and colleagues suggests that cytoprotective defenses may play a role in determining whether iron induces a necroinflammatory response [78]. These authors crossed *HFE^−/−^* mice with mice lacking the transcription factor nuclear factor erythroid-related factor-2 (NRF2), which regulates the expression of a large number of genes involved in protection against oxidative stress. Exogenous iron overload has been shown to induce several targets of NRF2, suggesting that this is an important mechanism of protection against iron [54,79,80,81]. *HFE^−/−^xNrf2^−/−^* mice showed increased hepatic fibrosis compared to either knockout alone, a finding that was strongly correlated with the number of necroinflammatory foci. Compromised cellular defenses are presumed to sensitize iron-loaded hepatocytes to necrosis, which then leads to generation of profibrogenic stimuli by macrophages when they ingest the necrotic, iron-loaded cells (Figure 1). Still, despite complete inactivation of a major orchestrator of protective responses to oxidative stress in the double knockouts, it is noteworthy that the magnitude of fibrosis induced by iron was relatively mild, with only a ≈2.5-fold increase in hepatic hydroxyproline content, even in elderly mice, which were the most severely affected [78]. Although it is reasonable to assume that fibrosis in these models results from interactions between iron, cellular defenses against oxidative stress, sideronecrosis, and inflammation, these findings may also be influenced by genetic differences in iron metabolism and susceptibility to inflammation and fibrosis among various strains of mice [72,82,83].

## 8. Interaction of Iron with Other Hepatic Insults

The combination of excess iron with other injurious agents has been reported to enhance hepatic injury and fibrosis in a number of cases. Chronic carbon tetrachloride (CCl_4_) administration is a commonly used model of liver injury characterized by hepatocyte necrosis; when CCl_4_ is administered chronically, it causes hepatic fibrosis. Using a low dose of CCl_4_ that would not by itself cause fibrosis, Mackinnon et al. demonstrated that the addition of iron or alcohol to the CCl_4_ regimen produced significant fibrosis [84]. Furthermore, the combination of all three agents (CCl_4_, iron, and alcohol) resulted in a synergistic increase in histological fibrosis score [84]. 

Another investigation found that a small amount of iron potentiated alcoholic liver injury in rats fed a high-fat diet [61]. In this experiment, the increase in hepatic iron following iron administration was relatively modest (less than 3-fold) and rats given iron without ethanol showed no increase in lipid peroxidation, ALT, AST, or histologic fibrosis. In contrast, the combination of iron and ethanol on the background of the high-fat diet significantly increased lipid peroxidation, ALT, and AST, as well as the histological fibrosis score. Interestingly, a small percentage of the animals given iron and ethanol even developed cirrhosis, which is a remarkable finding, since like iron overload models, animal models of alcoholic liver disease rarely develop progressive fibrosis [61,85]. 

Genetic models of iron overload in combination with additional stressors have shown similar results. For example, when *HFE*^−/−^ mice (with 2-fold increase in hepatic iron) were treated with ethanol, they developed histological fibrosis and increased mRNA expression of collagen-1 and αSMA [60]. Alcohol treatment was also associated with apoptosis and inflammation in the livers of the *HFE*^−/−^ mice, both features that were absent in mice that did not receive alcohol [60]. Similarly, *Hjv^−/−^* mice (with hepatic iron concentrations 14-fold higher than wild-type mice) fed a high-fat diet demonstrated increased hepatic α-SMA protein expression, although fibrosis was not potentiated in these livers as assessed by Sirius red staining [33]. 

In considering these various animal models, it is worth noting that even when a profibrogenic effect of iron has been shown, either alone or in combination with some other type of liver pathology, the resulting fibrosis has generally been mild and has thus required sensitive means to demonstrate and quantitate it (e.g., Sirius red staining, hydroxyproline quantitation, electron microscopy, (etc.). Progressive, histologically demonstrable fibrosis in response to iron loading in an animal model has rarely been described [86]. 

## 9. Iron as a Hormetic Agent 

Contrary to the commonly held view of iron as hepatotoxic, there are numerous examples of iron inducing protective responses in the liver, including the upregulation of ferritin, heme oxygenase-1, gamma glutamylcysteine synthetase, cysteine, thioredoxin1, NAD(P)H:quinone oxidoreductase, and tumor suppressor proteins [54,80,87,88].

Presumably as a result of its ability to induce protectants, iron administration has been shown to mitigate liver injury in response to some toxins. For example, Wang et al. observed that pretreatment with carbonyl iron dramatically attenuated biochemical and histologic evidence of liver injury following a single dose of carbon tetrachloride [89]. Similarly, Moon and associates demonstrated that peak ALT following a moderately toxic dose of acetaminophen was significantly lower in mice that had been pretreated with iron for 4 weeks compared to controls. Iron loading was also associated with a reduction in histologic evidence of injury and maintenance of glutathione levels following acetaminophen injection [90]. 

The ability of iron to confer protection has also been observed in models of liver injury that do not involve toxins. Kirsch et al. examined the effects of carbonyl iron on rats fed a methionine-choline deficient (MCD) diet as a model of non-alcoholic fatty liver disease [91]. These authors found that ALT levels were lower in rats fed the high-iron diet combined with the MCD compared to the animals given the MCD diet alone. Histological fibrosis was significantly more extensive in the MCD group compared to controls but was not significantly exacerbated by the combination of iron + MCD diet. Lipid peroxidation, as assessed by TBARS and by conjugated dienes, was remarkably decreased by the combination of iron and MCD compared to levels in the livers of the rats fed MCD diet alone, demonstrating a robust protective effect of iron [91]. Notably, these authors observed that iron overload was associated with a significant reduction in hepatic triglyceride on the MCD, confirming previous results that demonstrate that iron status influences lipid metabolism [92]. 

Another study demonstrated protective effects of iron in both wild-type and hemojuvelin knockout mice fed a high-fat diet (HFD). In wild type mice, the HFD was associated with significant increases in ALT and AST levels; this effect was abrogated in mice fed the HFD with a 2% carbonyl iron diet [33]. Despite very high hepatic iron concentrations (≈20,000 µg/g dry liver) *in Hjv^−/−^* mice fed the HFD plus iron, there was no histologically evident fibrosis; in fact, the livers of *Hjv^−/−^* mice fed the combination of HFD and iron had less steatosis and decreased expression of αSMA compared to *Hjv^−/−^* mice fed the HFD alone [33]. Collectively, these results demonstrate that iron can induce hormesis in rodent livers, conferring protection against subsequent toxic or metabolic injury. Whether similar phenomena occur in humans with iron overload is unknown. 

## 10. Iron-Induced Liver Injury: Hemochromatosis

Hereditary hemochromatosis (HH) is the prototypical condition linking excess iron to hepatotoxicity. Liver-related pathology (cirrhosis, hepatocellular carcinoma) has been reported to be an important cause of mortality in patients with HH [93]. Given this relationship, it may be surprising to learn that HH is a relatively uncommon cause of cirrhosis. Our understanding of this condition has evolved since 1996 when the mutations responsible for the most common form of HH were described [94]. In the years preceding the identification of the mutations, it was believed that homozygotes for this autosomal recessive condition invariably developed complications of iron overload, including cirrhosis, if not diagnosed and treated in a timely fashion. The assumption persisted despite the availability of evidence to the contrary, even in the pre-*HFE* era. For example, a study of over 5000 liver transplants done at 37 transplant centers in the United States (US) from 1981–1992 found that only ≈1% of transplants were performed for liver disease attributed to hemochromatosis, a remarkably low number for an ostensibly common cause of cirrhosis [95]. Similarly, a study of causes of death in the U.S. from 1979–1992 as recorded in an administrative database found that hemochromatosis was noted as an underlying or contributory cause of death at a much lower rate than expected based on the estimated prevalence of HH in the general population [96]. In the absence of a definitive marker to identify individuals with this condition, these types of data were presumed to reflect the “tip of the iceberg,” with the claim that many cases went undiagnosed due to the failure of physicians to recognize HH. 

Although the magnitude of the risk of end-organ damage in untreated HH remains a matter of debate, multiple studies in the years since the discovery of the HH mutations make it clear that disease manifestations in untreated HH homozygotes are considerably less common than was previously thought. The authors of a study that included ≈40,000 subjects estimated that less than 1% of those who were homozygous for the *HFE* C282Y mutation (the genotype associated with the disorder) manifested signs or symptoms attributable to HH [97]. Several studies from Europe found that the proportion of C282Y homozygotes was similar among elderly versus younger subjects, suggesting no adverse selection, i.e., early loss of HH homozygotes from the population, as would be expected if HH led to shortened life expectancy [98,99,100]. Consistent with these reports, low rates of biopsy-proven cirrhosis were found among newly-diagnosed HH patients identified via a population screening program from Norway, in a review of previously diagnosed HH cases from Wales, and in a screening study from North America [101,102,103]. 

Low rates of cirrhosis among C282Y homozygotes are attributable in part to low disease penetrance (i.e., lack of development of heavy iron loading) in a proportion of these individuals. Early detection and treatment likely also affect rates of cirrhosis by preventing the development of iron overload and its sequelae. These factors complicate the assessment of iron-induced hepatotoxicity among HH homozygotes as a group. Of particular relevance to this question is a study from Australia that reported iron overload-related disease in 28% of male HH homozygotes and <1% of female homozygotes [104]. Notably, this higher rate of disease manifestations (which included symptoms such as fatigue and use of arthritis medications, in addition to history of liver disease) was seen specifically in the group of HH homozygotes with significant iron burdens (defined as “heavy iron” on liver biopsy or serum ferritin >1000 g/L), i.e., subjects with penetrant disease with heavy iron burdens that had not been mitigated by treatment. The apparent discrepancy between these results and the others cited can be explained by the populations under study. It is not surprising that rates of iron overload-related disease were higher in a group of HH homozygotes with heavy iron burdens than among unselected HH homozygotes since, with the exception of arthropathy, the manifestations of HH occur in the setting of heavy iron overload. Indeed, ferritin levels exceeding 1000 µg/L are known to identify HH subjects at risk of cirrhosis [105]. Arguably the most intriguing observation from this work is that, even among HH homozygotes with heavy iron overload, only a minority had liver disease. This is compelling evidence in support of the concept that iron is less intrinsically hepatotoxic than is commonly assumed. 

Studies have shown that the risk of cirrhosis in HH is increased in the context of heavy alcohol use, hepatic steatosis, or polymorphisms in genes that sensitize to iron-related injury [106,107,108,109]. Consistent with these observations, we found that HH accounted for less than 0.05% of nearly 6000 cases of cirrhosis seen at our academic liver transplant center over a 10-year period [110]. In the majority of HH cases with cirrhosis, additional causes of liver injury were present in addition to iron overload (excess alcohol, chronic viral hepatitis, etc.). Only approximately 1 in 3000 cases of cirrhosis seen at our center was attributable solely to iron overload caused by HH, indicating that, while HH can indeed cause cirrhosis, this is an uncommon outcome. 

Phlebotomy effectively removes excess iron and is regarded as standard therapy for HH patients with evidence of expanded body iron stores. Studies predating the era of genetic testing provide evidence for a beneficial effect of phlebotomy [93], but this treatment has not been subjected to a randomized controlled trial. In patients who are diagnosed with HH but who do not have end-organ damage, such as hepatic fibrosis, phlebotomy is a prophylactic measure, done to prevent the development of organ damage in the future. Based on our current knowledge of this disease, it is reasonable to assume that phlebotomy is unnecessary in some proportion of these individuals who are not destined to develop end-organ damage. The difficulty is that there are currently no reliable means of identifying those who will progress to liver (or other organ) damage and are therefore most likely to benefit from phlebotomy. Conversely, removal of excess iron has been reported to improve fibrosis scores and reduce portal hypertension in a subset of HH patients with advanced liver fibrosis. Falize et al. observed fibrosis regression following phlebotomy in about half of their HH patients with advanced fibrosis (17/36, 47%) [111]. All of the subjects in that study had very heavy iron burdens initially, but the authors found that fibrosis regression was not strongly associated with adherence to treatment, nor was it related to the amount of iron removed [111]. Fracanzani et al. found that varices improved or completely reversed in 26% of patients with cirrhosis attributed to HH over a mean of 6 years of treatment [112]. A similar theme emerges from a study of β-thalassemia patients with hepatic fibrosis resulting from secondary iron overload who were treated with an oral iron chelator [113]. In contrast to the Falize study, most subjects in this study did not have advanced fibrosis at baseline. Nonetheless, reductions in fibrosis scores were seen in only ≈30% of the subjects after chelation and improvements in fibrosis were not strongly linked to decreases in liver iron [113]. The improvements effected by iron removal in these studies are not negligible, but it is noteworthy that removal of excess iron as a therapeutic intervention in these studies achieved a demonstrable benefit in less than half the cases, even in the absence of established cirrhosis. One might assume that even in the absence of fibrosis regression or improvement in portal hypertension, iron reduction may have prevented worsening of these parameters, but this is unproven. This is not to suggest that these conditions should not be treated with iron reduction, but rather to highlight the relative effectiveness of iron reduction in these specific conditions, in which excess iron plays an unquestioned role in disease pathogenesis. This provides necessary context for the discussion below on iron reduction as therapy for liver diseases which are not primarily caused by excess iron.

## 11. Abnormalities of Iron Metabolism in Non-Hemochromatosis Liver Disease

The data regarding the low prevalence of advanced liver disease resulting from HH notwithstanding, claims that iron is highly toxic to the liver are commonly encountered in the clinical literature. Referencing HH as the prototype, abnormalities of iron metabolism have been linked to progression of liver diseases caused by conditions other than HH. Dysregulation of iron metabolism in some cases has been attributed to heterozygosity for HH mutations; in other instances, the underlying abnormality of iron metabolism appears to be caused by the liver disease itself. The nexus between dysregulated iron metabolism and non-HH liver disease has been examined in greatest detail in chronic hepatitis C infection (HCV) and non-alcoholic steatohepatitis (NASH) [114,115]. This research indicates that 20–40% of HCV patients have alterations in iron metabolism as evidenced by elevated serum iron studies (serum iron, transferrin saturation, and/or ferritin levels), but only a minority of these individuals have excess hepatic iron deposition on liver biopsy [114,116,117,118]. Abnormalities of serum iron measures have been reported in variable proportions of NASH patients, with elevated transferrin saturation seen in up to one-quarter of patients and increased serum ferritin in half or more. As with HCV, increases in hepatic iron content are considerably less common than elevations of the serum iron markers, with hepatic iron deposition reported in one-third or less [119]. 

Despite the fact that these disturbances in iron metabolism (and bona fide hepatic iron accumulation) occur in a minority of patients with either HCV or NASH, one frequently encounters assertions that iron is integral to the pathogenesis of these conditions. However, the conclusion that iron is a key factor in the progression of either HCV or NASH based on the observation that iron metabolism is dysregulated in a subset of individuals with these conditions is unwarranted. First, this fails to account for the fact that the majority of subjects with these conditions do not have evidence of altered iron metabolism. Second, since bona fide hepatic iron overload is relatively uncommon among either HCV or NASH patients, the linkage between disease pathogenesis and iron is predicated on increased transferrin saturation and/or serum ferritin. The binding of iron to either of these proteins constrains its reactivity, so claims that high levels of transferrin saturation or ferritin are evidence for iron-driven Fenton chemistry, hydroxyl radical formation, and the like are speculative at best. Third, although serum iron levels are sometimes elevated in these conditions, an increase in transferrin saturation can also result from a decrease in transferrin. Transferrin is a negative acute-phase reactant, so its levels can be decreased by inflammation or liver injury per se [120]. Ferritin, on the other hand, is a well-known positive acute-phase reactant, whose levels are elevated by the same processes [121]. It should also be noted that serum ferritin has a subunit composition that distinguishes it from most types of tissue ferritin, and, unlike tissue ferritin, serum ferritin contains little iron [122]. Thus, alterations in the levels of these proteins are a reflection of the underlying hepatic inflammatory process rather than expansion of total body iron stores in most cases. Consistent with this notion, cure of HCV infection normalizes iron studies in the majority of hepatitis C patients who have elevated iron studies prior to antiviral treatment [Hasan and Brown, manuscript in preparation]. One could hypothesize that changes in the levels of these proteins are a marker for alterations in iron handling that may ultimately contribute to tissue damage, but it is difficult to implicate these parameters directly in iron-driven injury. Lastly, the proposed relationship between altered iron status and progressive liver injury is based on observations that patients with higher fibrosis scores are more likely to have evidence of altered iron metabolism than those with lower fibrosis scores [123,124,125,126]. Although these data are clearly insufficient to establish that dysregulation of iron metabolism is the cause of worsening fibrosis, they are commonly interpreted as demonstrating just that based on unsubstantiated assumptions regarding the behavior of iron (i.e., Fenton chemistry, hydroxyl radicals, etc.) discussed above. The converse interpretation, i.e., that worsening fibrosis causes dysregulation of iron metabolism, is supported by studies showing that hepcidin expression is reduced in the setting of advanced liver disease, indicating that dysregulation of iron metabolism is likely a consequence of worsening liver function [127,128]. It is conceivable that there is a positive feedback loop between these processes, with advanced liver disease causing altered iron metabolism, which in turn exacerbates the underlying liver disease; however, there is no direct evidence to support this chain of events at present.

Insofar as iron has been implicated in the pathogenesis of liver disease due to HCV and NASH, it is reasonable to expect that iron reduction might be a useful intervention in these conditions. Prior to the availability of highly efficacious treatment for HCV, there were multiple trials of iron reduction as a means of improving response rates in HCV patients treated with interferon-based therapy [129,130,131,132]. In aggregate, these studies showed that phlebotomy tended to lower liver enzyme levels but had negligible effect on rates of viral clearance in response to interferon. A meta-analysis of six of these trials subsequently showed a significant improvement in viral clearance with iron reduction, suggesting that these trials may have been inadequately powered [133]. This result is moot, however, given the current generation of highly effective HCV treatments. Long-term data concerning the effects of iron reduction on disease progression and HCC risk are scant, but a small trial from Japan showed beneficial effects on necroinflammation and fibrosis in HCV patients who were treated with iron reduction (phlebotomy + low-iron diet) over a 6-year period; the risk of HCC was decreased in HCV patients treated in this manner for up to 12 years [134,135]. It is important to note that in both of these studies, iron reduction was not intended merely to remove excess iron but to induce frank iron deficiency. The distinction between normalizing excess iron versus inducing iron deficiency is important and has potential consequences for understanding the means by which modulation of iron status may affect disease progression as discussed below.

Similar to the HCV trials, iron reduction has been reported to yield modest improvements in liver enzymes in NASH patients, but has otherwise not demonstrated consistent improvement in other parameters [136]. Although these results overall are not encouraging, most of the phlebotomy studies done to date have been relatively small and relatively short-term in nature, in addition to being heterogeneous in terms of study design, endpoints, etc. It is therefore possible that a subset of NASH patients might benefit from phlebotomy, or that longer-term iron reduction might be protective, but this remains to be seen. 

## 12. Iron Reduction Versus Iron Deficiency

Interpretation of the results of the iron reduction trials in HCV and NASH is complicated by the fact that the subjects in many of these trials were heterogeneous with respect to iron status. Removal of excess iron from subjects with expanded body iron stores as a therapeutic intervention is distinct from the induction of a state of overt iron deficiency in subjects with normal or increased iron stores at baseline. If restoration of iron stores to a normal level mitigates injury in the former case, it is reasonable to conclude that excess iron played a role in the injury. In contrast, mitigation of injury by the induction of iron deficiency is more difficult to interpret, since iron is required for numerous physiological processes. 

Induction of iron deficiency has been shown to modify inflammatory responses in a number of animal models. In experiments that go back nearly 4 decades, mild iron deficiency elicited by feeding an iron-deficient diet or by treatment with the iron chelator DFO was reported to suppress joint inflammation in several models of inflammatory arthritis [137,138,139]. More recently, Otogawa et al. demonstrated that rats fed an iron-deficient diet prior to thioacetamide (TAA) administration were protected from acute TAA hepatotoxicity, while rats that continued on a low-iron diet concurrent with chronic administration of TAA or following bile-duct ligation showed diminished fibrotic responses [140]. Darwish et al. showed that treatment with DFO abrogated the modest increase in hepatic iron content seen in concanavalin-induced fibrosis in rats and blunted concanavalin-induced increases in histological fibrosis as assessed using trichrome and α-SMA staining, and via hydroxyproline and TGFβ levels [141]. Concanavalin administration increased several markers of inflammation (e.g., IL-6 and interferon-γ), and these changes were also attenuated by DFO [141]. In the choline-deficient diet model of steatohepatitis, the oral iron chelator deferasirox decreased histological fibrosis as assessed by Sirius red staining, as well as expression of αSMA, TGFβ, and procollagen I [142]. These anti-inflammatory and anti-fibrotic effects of iron reduction warrant further study, but they do not prove that excess iron per se drives the underlying disease process, particularly in models of liver injury that are characterized by relatively small, if any, increases in iron stores. 

Under such circumstances, iron reduction may modulate injury responses in an indirect manner. For example, reduced availability of iron can limit proliferative responses as a consequence of the well-known role of iron in cell division [143]. Although the involvement of iron in cell division has been attributed to the iron dependence of ribonucleotide reductase, the enzyme that catalyzes the rate-limiting step in the biosynthesis of deoxyribonucleotides, recent research suggests a more expansive role for iron in DNA synthesis and repair [144]. Additionally, evidence from both animal and clinical studies indicates that iron deficiency impairs cell-mediated immune responses, at least in part as a result of suppression of T lymphocyte proliferation and reduced secretion of interferon-γ [145,146,147]. Innate immune responses are also modulated in response to iron deficiency, with diminished bactericidal activity of macrophages and reduced neutrophil myeloperoxidase activity [148]. Of particular relevance in chronic liver injury, iron is a cofactor for the prolyl and lysyl hydroxylases that catalyze the post-translational processing of collagen [71]. Thus, there are multiple mechanisms by which iron deficiency might indirectly ameliorate chronic liver injury, but these effects do not imply a specific causal role for iron in the injury process.

## 13. Conclusions

In this review, we provide a large body of evidence from animal experiments and clinical studies that challenges the notion that iron is intrinsically damaging to the liver. We show that, in animal models, excess iron typically causes a minimally histologically detectable fibrotic response that does not progress to cirrhosis. On the contrary, iron overload has protective effects in some models of hepatic injury. Further, we demonstrate that rodent livers are not unique in terms of their ability to tolerate high levels of iron without overt damage, as evidenced by the low rate of liver disease attributable to HH, even among *HFE* homozygotes with heavy iron overload. We discuss that while dysregulation of iron metabolism is sometimes observed in liver diseases such as chronic hepatitis C and nonalcoholic fatty liver disease, strong evidence that iron is an independent factor driving progression of these diseases is lacking, as is evidence of benefit from iron reduction. Based on the evidence provided here, we urge re-evaluation of the current paradigm that assumes that iron invariably plays a deleterious role in liver disease.

## Figures and Tables

**Figure 1 ijms-20-02132-f001:**
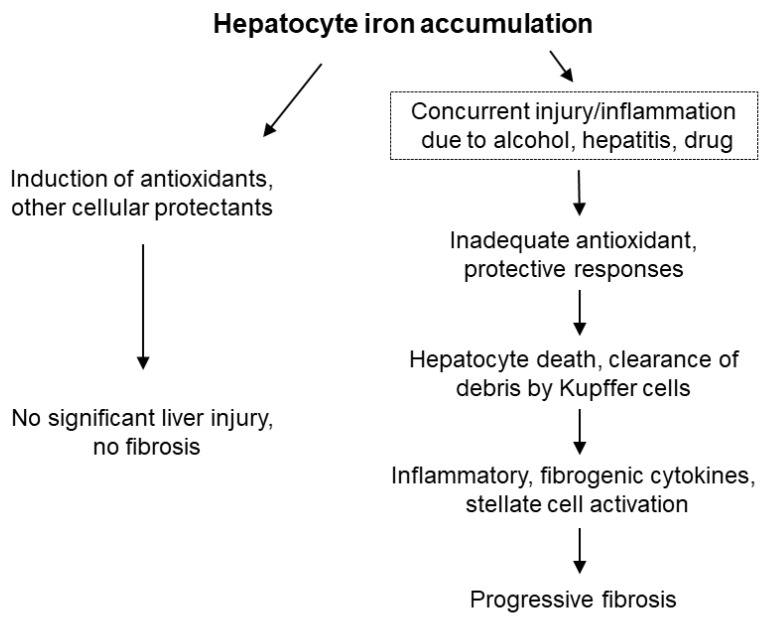
Potential outcomes of hepatocyte iron accumulation. In the setting of iron accumulation alone, iron elicits protective responses and minimal, if any, damage or fibrosis ensues. When iron accumulation occurs in the context of concurrent injury, the induction of protective responses may be impaired, leading to hepatocyte death. Clearance of debris from iron-laden hepatocytes by macrophages generates inflammatory and profibrogenic signals, resulting in progressive fibrosis.

**Table 1 ijms-20-02132-t001:** Animal models of chronic iron overload and degrees of fibrosis.

Reference	Method of Iron Administration	Animal Model	HIC (Fold Increase)	Degree of Fibrosis
Asare et al., 2006 [52]	2% carbonyl	Wistar rats	35	None
Brown et al., 2003 [54]	Iron-dextran	Sprague Dawley rats	75	None
Padda et al., 2015 [33]	Genetic	Hjv^−/−^ mice, 10 weeks old (C57BL/6 background)	14	None
Stål and Hultcrantz, 1993 [58]	3% carbonyl	Sprague Dawley rats	12	None
Stål et al., 1995 [59]	2.5–3% carbonyl	Wistar rats	14	None
Brissot et al., 1983 [53]	Intramuscular	Baboons	24–105	“Slight fibrosis”
Iancu et al., 1987 [69]	Intramuscular	^†^ Baboons	24–105	Detection of collagen fibrils via electron microscopy (EM)
Pietrangelo et al., 1990 [57]	3% carbonyl	Wistar rats	15	Mild periportal fibrosis via trichrome staining
Valerio and Peterson, 2000 [67]	0.2% ferrocene in diet 90 days, 0.04% 25 days	C57BL/6Ibg mice	15	Mild centrilobar fibrosis via trichrome staining
Roberts et al., 1993 [70]	2.5% carbonyl	Sprague Dawley rats	41	Collagen fibers via EM in rats with heaviest iron deposition
Weintraub et al., 1985 [71]	Iron-dextran	Albino rats	21	Collagen fibers via EM
Park et al., 1987 [66]	3% carbonyl	Sprague Dawley rats	54	“Mild to moderate”
Pietrangelo et al., 1995 [74]	Iron dextran	Gerbils	28	4.4-fold increase in liver collagen

Only models that increased hepatic iron concentration (HIC) greater than 10-fold are included in this table; other models are discussed in the text. The reference number is given in parentheses. ^†^ The same set of animals as described in Brissot et al., 1983 [53].

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
