# Peer review of "Iron-Induced Liver Injury: A Critical Reappraisal"

_ijms, 2019, doi:10.3390/ijms20092132_

Reviewer 1 Report

The manuscript by Bloomer and Brown is a highly comprehensive review that challenges the dominant view that iron is hepatotoxic. The manuscript is well structured and well written, but Fig.1 and Table 1, which are cited in the text, are missing. Other than that, I have only the following comments/criticisms:

1.       By citing both old and recent studies performed with animal (mostly rodent) models of iron overload, the manuscript provides convincing evidence that iron per se is poorly fibrogenic/cirrhotic, although it may enhance the toxicity of other toxic agents. On the other hand, the view of iron as a hormetic agent is more debatable, not least because it is supported by just a few animal studies. As for other pro-oxidants, it is not surprising that an iron excess increases the cellular/tissue antioxidant defenses. In some particular conditions, iron administration may have generated a mild oxidative stress that has primed the antioxidant defenses, thus protecting against a subsequent insult. However, this would hardly apply to situations of frank iron overload, and there is no evidence that this is the case with humans/patients.

2.       The statement that hereditary hemochromatosis (HH) is not currently a common cause of fibrosis/cirrhosis does not really exclude the possibility that iron is hepatotoxic. The question is more complex than that and readers should be made aware of this. A first reason why most carriers of the C282Y HFE mutation do not develop liver disease is the low disease penetrance. Not just the clinical penetrance, but also the biochemical penetrance, meaning that despite the genotype a lot of these individuals do not develop massive iron overload. Without so they should not really be considered HH patients (see Porto G et al. 2016 ‘EMQN best practice guidelines for the molecular genetic diagnosis of hereditary hemochromatosis (HH)’ Eur J Hum Genet). Most epidemiological studies have not distinguished C282Y HFE homozygous from true HFE-hemochromatosis patients. A second and very important reason is the fact that for a few decades HH patients have been diagnosed early and treated with phlebotomies. In many cases, the treatment starts well before the appearance of clinical symptoms (preventive phlebotomies). This would be expected to prevent the development of cirrhosis, and may partly explain the very low proportion of cirrhosis cases attributed to HH in the authors academic liver transplant centre (ref 110). Moreover, and as the authors rightly point out, iron removal through phlebotomy has proven useful to improve liver injury in HH patients with advanced fibrosis.

Author Response

Response to Reviewer 1

We thank the Reviewer for his/her helpful comments.  We do not know why Fig 1 and Table 1 were not included in the documents for the initial review but will try to rectify this problem on our resubmission. 

1.  The Reviewer expresses doubt that iron can be considered a hormetic agent based on the “few animal studies,” indicates that it is unsurprising that the mild oxidative stress generated by iron administration induced upregulation of antioxidant defenses and further states that this would “hardly apply to situations of frank iron overload and there is no evidence that this is the case with humans/patients.”

First, we wish to point out that our objectives in this review were to consider the hepatotoxicity of iron broadly, hence the separate treatment of animal models and human diseases.  Although they may be relatively few in number, the animal studies cited in this section show that iron pretreatment can induce remarkable protective effects in some circumstances.  For example, the study of Wang et al (reference 89) demonstrated dramatic decreases in aminotransferases in rats pretreated with carbonyl iron prior to CCl4 administration (mean ALT 206 vs 1856 in rats given CCl4 without prior iron; histologic injury was also substantially reduced).  This certainly seems to fit the definition of hormesis.  We agree that it is unknown whether iron overload induces hormesis in humans, but this does not diminishes the potential importance of this phenomenon to researchers who may use rodent models to investigate interactions between iron and other injurious agents.  To our knowledge, the question of iron-induced hormesis in humans has never been investigated.  We concede the Reviewer’s point regarding the lack of information on this point and have noted this in the revised manuscript.

Regarding the Reviewer’s other comments on this topic, it is not clear what the Reviewer meant by “frank iron overload.”   The hepatic iron concentrations achieved in the study that we cited showing that Hjv-/- mice fed carbonyl iron were protected from steatohepatitis were very high  (~20,000 mg/g dry weight).  We have shown induction of the glutathione synthetic machinery as well as other putative cellular protectants, including a1-acid glycoprotein (the major rat acute phase reactant) and metallothioneins 1 and 2, in rat livers with >50-fold increases in hepatic iron concentrations compared to untreated rat livers (reference 80).  This indicates that induction of protective responses unequivocally can occur in the context of heavy iron loading (at least in rodent models) and that the protectants induced are not limited to traditional antioxidants.  Again, whether similar events take place in human livers is unknown and we do not claim otherwise.

2.  The Reviewer states that the lack of significant liver disease in most C282Y homozygotes does not exclude the possibility that iron is hepatotoxic, since many C282Y homozygotes do not develop massive iron overload (low penetrance) and in other cases, early detection and treatment of the disease may have prevented both the development of heavy iron overload and the resulting liver damage.  We agree that these factors contribute to the overall low rate of advanced liver disease attributable to HH and have added text to this section to highlight this point.  We also agree with the Reviewer that labelling all C282Y homozygotes as having “hemochromatosis” regardless of whether they have significant iron burdens is incorrect and confusing.  However, our main objective in this section was to discuss how it came to be that this previously ostensibly common cause of liver disease is now recognized as relatively uncommon; the issue of proper nomenclature for non-expressing C282Y homozygotes is important but beyond the scope of this already lengthy review. However, contrary to the Reviewer’s statement that low rates of penetrance and early treatment make it difficult to exclude that iron is hepatotoxic, we would point to the Australian study (cited as reference 104) which showed low rates of advanced liver disease even among C282Y homozygotes selected for heavy iron overload (i.e., subjects with penetrant disease, and presumably no prior treatment), which speaks directly to the question of hepatotoxicity.

Reviewer 2 Report

The review manuscript describes the recent concepts on iron-induced liver injury. The paper was well organized and very informative to researchers in this field.

One minor recommendation. Could you consider to add scheme(s) for some concept(s) important in this review? It would be helpful to readers to understand easily the points of this review.

Author Response

Response to Reviewer 2

1.  We thank the Reviewer for his/her supportive comments.  We agree that a conceptual schema is helpful for understanding the concepts discussed in this review.  We provided such a schema in Figure 1, which for some reason was not included with the initial version of the manuscript.  We hope that it is included on the second round of reviews and that the Reviewer finds it helpful.

Round  2

Reviewer 1 Report

The authors have incorporated my comments in the manuscript, so I have nothing against its publication, except for the fact that, for some reason, Table 1 and Figure 1 are still missing.

Author Response

We have again revised the MS and hope that the Table and Figure are finally attached!